# Spatiotemporal Evolution and Driving Forces of Vegetation Cover in the Urumqi River Basin

**DOI:** 10.3390/ijerph192215323

**Published:** 2022-11-19

**Authors:** Azimatjan Mamattursun, Han Yang, Kamila Ablikim, Nurbiya Obulhasan

**Affiliations:** 1Institute of Geography and Tourism, Xinjiang Normal University, Urumqi 830054, China; 2Xinjiang Key Laboratory of Lake Environment and Resources in Arid Zone, Xinjiang Normal University, Urumqi 830054, China; 3School of Public Management, Xinjiang Agricultural University, Urumqi 830052, China

**Keywords:** fractional vegetation coverage, trend analysis, OPGD, arid area, Urumqi River basin

## Abstract

It is important to determine long-term changes in vegetation cover, and the associated driving forces, to better understand the natural and human-induced factors affecting vegetation growth. We calculated the fractional vegetation coverage (FVC) of the Urumqi River basin and selected seven natural factors (the clay and sand contents of surface soils, elevation, aspect, slope, precipitation and temperature) and one human factor (land use type). We then used the Sen–Man–Kendall method to calculate the changing trend of the FVC from 2000 to 2020. We used the optimal parameters-based geographical detector (OPGD) model to quantitatively analyze the influence of each factor on the change in vegetation coverage in the basin. The FVC of the Urumqi River basin fluctuated from 2000 to 2020, with average values between 0.22 and 0.33. The areas with no and low vegetation coverage accounted for two-thirds of the total area, whereas the areas with a medium, medium–high and high FVC accounted for one-third of the total area. The upper reaches of the river basin are glacial and forest areas with no vegetation coverage and a high FVC. The middle reaches are concentrated in areas of urban construction with a medium FVC. The lower reaches are in unstable farmland with a medium and high FVC and deserts with a low FVC and no vegetation. From the perspective of the change trend, the areas with an improved FVC accounted for 62.54% of the basin, stable areas accounted for 5.66% and degraded areas accounted for 31.8%. The FVC showed an increasing trend in the study area. The improvement was mainly in the areas of urban construction and desert. Degradation occurred in the high-elevation areas, whereas the transitional zone was unchanged. The analysis of driving forces showed that the human factor explained more of the changes in the FVC than the natural factors in the order: land use type (0.244) > temperature (0.216) > elevation (0.205) > soil clay content (0.172) > precipitation (0.163) > soil sand content (0.138) > slope (0.059) > aspect (0.014). Apart from aspect, the explanatory power (*Q* value) of the interaction of each factor was higher than that of the single factor. Risk detection showed that each factor had an interval in which the change in the FVC was inhibited or promoted. The optimum elevation interval of the study area was 1300–2700 m and the greatest inhibition of the FVC was seen above 3540 m. Too much or too little precipitation inhibited vegetation coverage.

## 1. Introduction

Vegetation is an integral part of the global ecosystem and is vital in the cycling of both matter and energy. Surface vegetation has essential functions in an ecosystem, such as carbon sequestration, the control of soil loss and the retention of pollutants. Surface vegetation undergoes interannual and seasonal changes and is sensitive to changes in the atmosphere, water and soil [1,2,3]. The fractional vegetation coverage (FVC) reflects both directly and indirectly the properties of the underlying surface and is a product of regional climate change and human activity. It is an important indicator of the quality of the regional ecological environment [4] and is affected by the topography (elevation and slope), precipitation, temperature, soil characteristics and the distribution of human populations [5,6,7]. It is therefore important to determine the long-term changes in the regional FVC to better understand the natural and human-induced factors affecting vegetation growth.

The acquisition and application of images with a high spatiotemporal resolution have been continuously improved with the development of remote sensing technology. Remote sensing measurements and statistical models are necessary for the large-scale, long-term monitoring of the FVC and analysis of the factors driving the growth of vegetation [8,9,10]. Previous studies have used medium- and low-resolution long-term vegetation index products, such as those from the Moderate Resolution Imaging Spectroradiometer (MODIS), to analyze the FVC on a large scale [11,12]. However, the use of data from medium- or low-resolution products will underestimate the FVC in arid areas where there are problems with continuity and mixed-pixel images [13,14,15,16]. The use of medium- to high-resolution images from the Landsat series of satellites at the basin scale reduces the errors caused by the spatial resolution of the data [17].

Linear regression and logistics regression have both been used to conduct statistical analysis of the FVC and its driving factors, but these methods ignore spatial heterogeneity. The geographical detector is a statistical method that can be used to analyze spatial heterogeneity based on the influence of independent variables on dependent variables and the similarity of spatial distributions, determined by calculating the spatial variance. This method can be used in environmental remote sensing to reveal the geographical differentiation of influencing factors (e.g., climate, vegetation, topography, terrain and soils) [18]. The geographical detector method has been used to analyze the factors driving the FVC—for example, a 15-year study of changes in the FVC in Sichuan Province showed that the topography could either promote or inhibit the FVC in the arid area of Dunhuang, whereas the temperature, wind speed and relative humidity had more significant roles [19,20]. The explanatory power of human factors was generally lower than that of natural factors [21]. The explanatory power of the interaction between factors on the FVC was increased compared with the explanatory power of a single factor on the FVC [21,22,23,24]. However, phenomena such as the subjective discretization of the driving factor data and the monitoring of spatial heterogeneity between changes in vegetation cover and single-period driving factors were not considered in previous studies.

There is a lack of accurate quantitative evaluations of the factors driving long-term changes in the FVC. In May 2020, optimal parameters based on the spatial data discretization method had been added to the original geographical detector. This method can improve the overall ability of the analysis of spatial hierarchical heterogeneity, such as spatial pattern analysis and spatial factor detection, to conduct more accurate, flexible and efficient spatial heterogeneity analysis [25].

The Urumqi River basin, located in the hinterland of the Eurasian continent, is a typical arid basin. It is home to major cities along the Silk Road Economic Belt in Xinjiang, including Urumqi City, Wujiaqu City and Changji Hui Autonomous Prefecture. It is important to analyze both the temporal and spatial distribution of changes in the FVC in this basin and their driving factors to understand differences in the quality of the ecological environment of the basin. We used remote sensing images of the Urumqi River basin obtained from Landsat satellites in each month of June–August in the time period 2000–2020 and the Normalized Differential Vegetation Index (NDVI) to monitor the spatiotemporal changes in the FVC of the basin. The Sen–Man–Kendall trend analysis method was used to evaluate the annual change trend. The driving forces of the FVC were quantitively analyzed based on the OPGD model with the aim of providing a scientific basis for the protection of the ecological environment.

## 2. Materials and Methods

### 2.1. Study Area

The Urumqi River basin is located in the hinterland of Eurasia, on the northern slopes of the Tianshan Mountains and the southern edge of the Junggar Basin (Figure 1). The river starts from the Tianshan Mountains in the south, reaches Dongdaohaizi on the southern margin of the GurbanTonggut Desert in the north, meets the Toutun River basin in the west, and reaches the Chaiwopu Depression in the east. The Urumqi River is 214 km long and the basin has a total area of about 6648.51 km^2^, with mountains, basins, plains and hills. The difference in elevation between the north and south of the basin is 4000 m and the vertical zones of soil and vegetation vary significantly. The basin is in a region with a mid-temperate dry continental climate, with low precipitation, a significant difference in temperature between the seasons, and long winters.

The vegetation of the basin belongs to 12 families and 35 species (Appendix A), mainly cold-resistant, drought-tolerant, and alkali-tolerant. The alpine region above 2000 m in elevation is dominated by *Picea schrenkiana*, and the subalpine region is distributed with different species of grasses; the middle reaches are covered with alkali-tolerant vegetation and various crops and fruit trees; the downstream desert region is distributed with *Haloxylon ammodendron* (Appendix A).

### 2.2. Data

#### 2.2.1. Landsat Series Satellite Images

The Landsat satellites have made important contributions to ecology, environmental monitoring, resource management, urban planning and disaster prediction, since their launch in July 1972. The correlation between the band data obtained by the Thematic Mapper (TM) onboard the Landsat satellites and the NDVI has been used to establish a band combination model to retrieve information about land-based vegetation [26]. The detection of vegetation with Enhanced TM Plus (ETM+) source data ensured the continuity of data before the sensors were damaged in 2003 [27,28]. The Landsat 8 satellite, launched in 2013, uses a more advanced technological design, which has improved its performance and reliability. The Operational Land Imager sensor onboard Landsat 8 not only narrows the spectral range of the original full-color band to make the contrast between vegetation and non-vegetation on the full-color image more significant, but also eliminates the effect of the absorption of water vapor at 0.825 μm and has a higher accuracy during the extraction of information [29]. We therefore selected one pair image of TM and Operational Land Imager images from the US Geological Survey website (http://earthexplorer.usgs.gov, accessed on 17 September 2022) for June–August of each year in the time period 2000–2020 to calculate the NDVI. The images have a spatial resolution of 30 m and strip numbers of 143-029 and 143-030 (see Appendix A); the cloud cover was <10%.

#### 2.2.2. Soil Sharacteristics

The 2010 Soil Featured Data Set provided by the National Qinghai–Tibetan Plateau Science Data Center is based on the Second National Soil Survey of China 1:1,000,000 soil map and 8595 soil section views. This dataset uses data from the US Ministry of Agriculture (USDA) China Land and Climate Simulation Standards to develop a multi-layer soil particle size distribution dataset. The distribution of the sand, powder and clay content of soils is derived using the polygonal link method, combining the distance between the soil section and the map polygon, the sample size of the section surface, and soil classification information (http://data.tpdc.ac.cn/en/data/3d2b6faf-2489-4770-b42e-84b392a15b07/?q=2010%20soil accessed on 17 September 2022). The resolution of the dataset is 1 km and it can be used for land and climate modeling in the study region [30]. We used the clay and sand content data for surface soils.

#### 2.2.3. Terrain Data

The Shuttle Radar Topography Mission (SRTM) is a joint mission between the National Aeronautics and Space Administration (NASA) and the Department of Defense National Imagery and Mapping Agency (NIMA). After more than two years of processing, the radar image data were released as a digital elevation model [31,32]. A large amount of information (e.g., the slope and aspect of basins) can be extracted through the SRTM 30-m dataset. It has a wide range of applications in surveying and mapping, hydrology, meteorology, geomorphology, geology, soil science, engineering construction, communications, military applications, national economics and defense construction, as well as in the humanities and natural sciences (https://srtm.csi.cgiar.org/ accessed on 17 September 2022). We obtained elevation, slope and aspect information from the SRTM 30-m data. Based on the slope angle, there are nine aspects: flat (<0°), north (0–22.5°, 337.5–360°), northeast (22.5–67.5°), east (67.5–112.5°), southeast (112.5–157.5°), south (157.5–202.5°), southwest (202.5–247.5°), west (247.5–292.5°) and northwest (292.5–337.5°).

#### 2.2.4. Meteorological Data

The 44 MODIS products can be divided into four unique data products: atmosphere, land, ice and snow, and marine. The MODIS/Terra Land Surface Temperature/Emissivity 8-DAY L3 Global 1km SIN Grid (MOD11A2.006) is a level 3 land product with a resolution of 1 km. Its pixel value is the average value of the surface temperature of the eight-day MOD11A1 product. Based on the 1B data, the product is corrected to the edge distortion generated by the remote sensor imaging process. It provides information such as the day and night surface temperatures, related quality indicators, the observation time and the sideline angle (https://ladsweb.modaps.eosdis.nasa.gov/search accessed on 17 September 2022) We used the average annual surface temperature for the time period 2000–2020.

The spatial resolution of China’s monthly precipitation data is 0.0083333° (about 1 km) for the time period 1901–2020 (http://data.tpdc.ac.cn/en/data/b5f99f9b-77db-4488-99db-6e21e4c34bc6/?q=precipitation%201901- accessed on 17 September 2022). This dataset is based on the global 0.5° climate dataset released by the Climate Research Unit and the global high-resolution climate dataset released by WorldClim, which is generated in China through the Delta spatial reduction scheme. Data from 496 independent weather observation stations are used for verification and the verification results are credible [33,34,35,36]. We used the monthly data for the time period 2000–2022 to obtain the cumulative annual precipitation.

#### 2.2.5. Land Use Data

China’s 30-m annual China Land Cover DataSet is based on 300,000 Landsat images, 5463 independent reference samples and visually interpreted samples, with an overall accuracy of 79.31% (http://irsip.whu.edu.cn/resources/CLCD.php accessed on 17 September 2022). It reflects China’s rapid urbanization and ecological engineering projects and reveals the impact of human activities on the regional land surface in the context of climate change [37]. We reclassified the original nine land use types into six types: farmland, forest, grass, water, construction and unused.

All images were resampled to 30 m resolution in order to complete the drive factor analysis.

### 2.3. Methods

#### 2.3.1. Fractional Vegetation Coverage

Gutman and Ignatov (1998) established a semi-empirical relationship between the *FVC* and the *NDVI* and proposed a dense MOSAIC pixel model to calculate vegetation coverage using the *NDVI* [38]. The *NDVI* is widely used and has a high correlation with the vegetation coverage and only small errors [39]. The dimidiate pixel model was used to invert *FVC*. The dimidiate pixel model assumes that a pixel includes both vegetation and soil. The formula is as follows:(1)$$\mathrm{FVC}=\frac{{\mathrm{NDVI}\;-\;\mathrm{NDVI}}_{\mathrm{soil}}}{{\mathrm{NDVI}}_{\mathrm{veg}}\;{-\;\mathrm{NDVI}}_{\mathrm{soil}}}$$
where *FVC* represents the fractional vegetation coverage, and *NDVI_veg_* and *NDVI_soil_* represent the *NDVI* values of areas with a high and low *FVC*, respectively. The maximum and minimum values of the *NDVI* are 99.5 and 0.5% of the pixel distribution. Finally, the values of *NDVI_veg_* and *NDVI_soil_* were determined, respectively (Suplementary Appendix A). We divided the *FVC* into five categories: no vegetation coverage (No__*FVC*_) (0–0.1), Low__*FVC*_ (0.1–0.3), Medium___*_FVC_* (0.3–0.6), Medium_High__*FVC*_ (0.6–0.8) and High___*_FVC_* (0.8–1).

#### 2.3.2. Theil–Sen Median Trend Analysis and the Mann–Kendall Test

The Theil–Sen median trend analysis and Mann–Kendall test can be combined to judge long-term trends in data and have been used in the long-term sequence analysis of vegetation [40,41,42,43]. The Theil–Sen median is a stable trend calculation method for nonparametric statistics and can reduce the influence of outliers [44]. The Theil–Sen median trend can be used to calculate the median slope of [*n*(*n* − 1)/2] data combinations as follows:(2)$$S_{\mathrm{FVC}}=\mathrm{Median}\frac{{\;(\mathrm{FVC}\;}_{j\;}{-\;\mathrm{FVC}}_{i})\;}{j\;-\;i}\;,\;2000\;\leq \;i\;<\;j\;\leq \;2020$$
where *S_FVC_* > 0 indicates that the area *FVC* is improving over time and *S_FVC_* < 0 means that the *FVC* is degrading over time.

The advantages of the Mann–Kendall test are that it does not need the data to follow a certain distribution, has a strong resistance to data errors and has a relatively solid statistical theoretical basis for testing the significance level, making the results more scientific and credible [45,46]. Its calculation is as follows:(3){FVC_i },i=2000, 2001, …, 2020
(4)$$\{\begin{array}{c} \frac{S\;-\;1}{\sqrt{s(S)}},\;S>0\\ 0\;,\;S=0\\ \frac{S+1}{\sqrt{s(S)}},\;S<0\; \end{array}\;,\;S=\overset{n-1}{\underset{j=1}{\sum }}\overset{n}{\underset{i=j+1}{\sum }}\mathrm{sgn}\left({\mathrm{FVC}}_{j}-{\;\mathrm{FVC}}_{i}\right)\;$$
(5)$$\mathrm{sgn}\left({\mathrm{FVC}}_{j}-{\;\mathrm{FVC}}_{i}\right)=\{\begin{array}{c} {1,\;\mathrm{FVC}}_{j}-{\;\mathrm{FVC}}_{i}\;>0\\ {0,\;\mathrm{FVC}}_{j}-{\;\mathrm{FVC}}_{i}=0\;,\\ {-1,\;\mathrm{FVC}}_{j}-{\;\mathrm{FVC}}_{i}\;<0\; \end{array}\;s(S)=\frac{n\left(n\;-\;1\right)\left(2n+5\right)}{18}$$
where *S_FVC_* > 0 indicates an increasing trend, *S_FVC_* < 0 indicates a downward trend and *n* is the annual span (equal to 20). *FVC_j_* and *FVC_i_* represent the *FVC* values of year *j* and year *i*; the statistic *Z* will be (−∞, +∞). At a given significance level α, when |*Z*| > *υ*_1_ − _α_/2, the changes in the trend are considered significant. We judged the significance of the *FVC* time series change trend at the 0.05 confidence level.

The change in the *FVC* in the basin was graded and evaluated according to the *S_FVC_* value of the regression equation to better evaluate the change in the *FVC* in the Urumqi River basin. The *FVC* was divided into five grades: decrease, slight decrease, stable, slight increase and significant increase.

#### 2.3.3. OPGD Model

The geographical detector uses quantitative and qualitative data to statistically monitor the spatial heterogeneity of research objects. It solves the subjective selectivity of discretization and evaluates the research object scientifically [18]. The OPGD model includes four parts: factor, interaction, risk and ecological. In this study, we discreted to each continuous variable factor and performed factor detection, interaction detection and risk detection.
Factor detection calculates the degree of influence *Q* of each factor on the *FVC*. The larger the value of *Q*, the greater the influence on the *FVC*. The formula is as follows:(6)$$Q=1-\frac{\sum_{h=1}^{l}N_{h}\sigma_{h}^{2}}{N\sigma^{2}}=1-\frac{SSW}{SST},SSW=\overset{l}{\underset{h=1}{\sum }}N_{h}\sigma_{h}^{2},SST=N\sigma^{2}$$
where *Q* is the spatial differentiation index, *h* = 1,…, *L* is the stratification of the attribute *FVC* or natural and human factors—that is, classification or partition; *N_h_* and *N* are the number of units in a particular layer and the whole area, respectively; $\sigma_{h}^{2}$ and $\sigma^{2}$ are the variances of the *FVC* values for layer *h* and the whole region, respectively; and *SSW* and *SST* are the sum of the intra-layer variances and the total variance of the whole area. The larger the value of *Q*, the greater the spatial difference in the *FVC*. If the hierarchy consists of factor *X*, the more explanatory the dependent variable *X* is for the attribute *FVC*.Interaction detection represents the interaction between different influencing factors. It compares the sum of single-factor *Q* values, double-factor *Q* values and double-factor interactions and evaluates whether the factors X1 and X2 increase or decrease the predictive power of the dependent variable *FVC*—that is, the main comparison factors *Q* (X1), *Q* (X1) + Q (X2) and *Q* (X1 ∩ X2).Risk detection is conducted by calculating the average *FVC* value of the subregion of the influencing factors for the statistical significance test. In each area, the larger the average value of *FVC*, the more suitable the sub-area of the influencing factor is for vegetation growth, which can be used to judge the appropriate range or type of each influencing factor. The expression formula is:(7)$$t=\frac{{\bar{Y}}_{h=1}-{\bar{Y}}_{h=2}}{{\left[\frac{Var\left(Y_{h=1}\right)}{n_{h=1}}+\frac{Var\left(Y_{h=2}\right)}{n_{h=2}}\right]}^{\frac{1}{2}}}$$
where ${\bar{Y}}_{h}$ is the average value of the *FVC* attribute in the *H* zone; $n_{h=1}$ is the number of samples in the *H* partition and Var is the variance.


## 3. Results

### 3.1. Long-Term Changes in the FVC

The changes in vegetation coverage in the research area between 2000 and 2020 showed irregular fluctuations (Figure 2). The FVC_mean_ (FVC mean value of all pixels in the basin) fluctuated between 0.22 and 0.33, with the lowest value of 0.23 in 2001 and the highest value of 0.32 in 2017. The FVC_mean_ increased and decreased significantly year-by-year from 2001 to 2004. During these three years, the FVC_mean_ changed significantly from −29 to 47%. The peak increase of 46.8% was in 2001–2002, whereas the smallest change of −28.4% occurred in 2002–2003. There was stable growth after 2004, except in 2009 (17.1%) and 2018 (−15.6%). The FVC_mean_ of the Urumqi River basin has therefore shown a slow growth trend since 2000, mainly in the range −10 to 10%.

Figure 3 shows the grading system used for the changes in the FVC in the research area in the time period 2000–2020. The study area was defined as having mainly No__FVC_ and Low__FVC_. The area of Medium__FVC_ was stable, whereas the areas of Medium_High__FVC_ and High__FVC_ decreased. The No__FVC_ and Low__FVC_ areas accounted for two-thirds of the total basin area (about 4000 km^2^). The average proportion of the No__FVC_ area in the total basin area was 19.16% (about 1303.34 km^2^), reaching 2002.68 km^2^ in 2001 and 1966.47 km^2^ in 2007. It then tended to decrease year-by-year in a small floating range. In 2020, the area was reduced to 796.88 km^2^. The average proportion of Low__FVC_ areas in the total basin area was 42.18% (about 2803.51 km^2^), reaching 45.65, 50.98 and 55.35% in 2001, 2003 and 2008, respectively, whereas it was much lower than the average value in other years. In the last decade, 2013, 2015 and 2017 were lower than the average value, whereas the rest of the years tended to increase, being close to, or exceeding, the average value, reaching the highest value of 48.57% in 2020 (about 3328.05 km^2^). The Medium__FVC_, Medium_High__FVC_ and High__FVC_ areas were highly stable at about 38.21% (about 2539.69 km^2^). The lowest proportions were 24.22% in 2001 and 25.65% in 2003; the other years were close to or above the average, with the highest proportion of 44.99% in 2009 (about 2990.26 km^2^).

The FVC in the study area showed particular distribution patterns in the basin’s upper, middle and lower reaches (Figure 4). The areas with No__FVC_ or Low__FVC_ were mainly distributed near Glacier No. 1 in the upper reaches of the basin, the buffer zone near the water source protection land of the Wulapuo Reservoir, around the built-up area in the middle reaches, and in the desert area with a poor natural environment in the lower reaches in the Xiaohaizi Desert. The Medium__FVC_ areas were mainly scattered in the middle and lower reaches, forming a buffer zone between the Medium_High__FVC_ and High__FVC_ areas and the No__FVC_ and Low__FVC_ areas. The Medium_High__FVC_ and High__FVC_ areas were distributed in the southwest, east and northwest, and surrounded the entire center of the basin. The distribution of the FVC in the study area was stable from 2000 to 2020. However, the vegetation coverage in the southwest and east of the basin shifted from High__FVC_ to Medium__FVC_ to Medium_High__FVC_, decreasing continuously.

### 3.2. Effects of Different Land Use Types on the FVC

Human activities can lead to changes in vegetation cover. Some land use types are more strongly affected by human disturbance, which may explain changes in the FVC. The area ratios of different FVC values in the basin were analyzed by considering the distribution of different land use types from 2000 to 2019 (Figure 5). Cultivated land showed a cyclical pattern in the ratio of different FVC values as a result of its vulnerability to human activities: the Medium__FVC_ and Low__FVC_ areas on average accounted for 66.46% of the total area of cultivated land. The medium vegetation cover reached the highest regional value of 42.89% of the area of same vegetation cover in the basin in 2015, about 423.26 km^2^; the low vegetation cover reached the highest value of 84.90% of the same vegetation cover in the basin in 2008, about 974.16 km^2^. In general, medium and low vegetation cover was dominant in the study area.

Grassland made an important contribution to the change in vegetation cover in the basin: the average percentages of the area covered by grassland, from no vegetation cover to high vegetation cover, as a proportion of the area of the basin during 2000–2019, were 43.86, 65.69, 61.74, 56.78 and 37.71%, respectively. The degradation or improvement of grassland directly influenced the level of vegetation cover in the basin. The two land use types of forest land and unused land changed from no vegetation cover to low and medium vegetation cover in each degree of vegetation cover, except for some abnormal values. These two land use types were generally green.

There were also some rebound situations. Over the 20-year study period, water bodies with vegetation accounted for 8.62% (7.22 km^2^) of the total area of water bodies, with no obvious transformation trend. The vegetation coverage in water bodies was mainly attributed to three aspects: the classification accuracy of land use type; the growth of aquatic plants; and the chlorophyll content of water bodies. The area of impervious surfaces (construction land) in 2019 had increased by about 86.12% (199.12 km^2^) compared with 2000. The average area covered by vegetation accounted for 64.91% of the total impervious surface area, reaching 78.71% in 2019, indicating that the cities were generally green. Impervious surfaces (construction land) with low and medium vegetation cover showed a linear growth trend with slopes of 5.52 and 5.86, respectively, and *R*^2^ values of 0.78 and 0.88, respectively. Human activities in the basin therefore had a positive effect on vegetation growth in residential areas. However, the vegetation cover of impervious surfaces (construction land) showed a continuously increasing trend, similar to a previous study on the effects of global urbanization on vegetation cover [47].

### 3.3. Long-Term Trends in the FVC

The FVC of the whole basin changed from 2000 to 2020. The area with changes in the FVC accounted for 94.34% of the total basin area (about 6272.20 km^2^) and 31.02% had a significant change. The area with an increasing trend of vegetation coverage accounted for 62.54% of the total basin area (about 4157.98 km^2^), more than half of the total area. The areas with different FVC levels were analyzed statistically to determine the trend in the FVC (Table 1). In total, 26.15% of the areas showed an increase in the FVC and 36.39% showed a slight increase (about 1738.59 and 2419.39 km^2^, respectively). The areas with a decrease in the FVC accounted for 31.8% of the total area of the basin (about 2114.22 km^2^), whereas 26.93% showed a small decrease and 4.87% showed a large decrease in the FVC.

The trend map of the FVC in the Urumqi River basin (Figure 6) shows that the areas with decreasing and increasing FVC values have prominent spatial characteristics. The decreasing area was concentrated in the southwest of the upper reaches of the basin and in the east and northwest of the middle reaches of the basin. The primary vegetation types were woodland, grassland and cultivated land. The area with a severe decrease in the FVC was distributed in the periphery of the built-up area of the city in the middle reaches of the river basin. The area with an increase in the FVC was mainly concentrated in the middle reaches of the river basin near the central city of Urumqi and the lower reaches in the Xiaohaizi Desert adjacent to Changji Prefecture and Wujiaqu City. Other areas with an increase in the FVC were scattered in the basin’s upper, middle and lower reaches. The area with an increase in the FVC was remarkably correlated with the direction of expansion of the urban area of Urumqi. The areas with an apparent increase were mainly concentrated in older urban areas (e.g., Tianshan District, Shaibak District, Xinshi District and Shuimogou District), whereas the areas with a slight increase were distributed in the north of the city (e.g., the Hemaquan, Daxue Road and Xishan areas).

### 3.4. OPGD Model in FVC

#### 3.4.1. Factor and Interaction Detectors

The *Q* values of the eight impact factors all passed the significance test. The average explanatory power from 2000 to 2020 was, in descending order: land use (0.244) > temperature (0.216) > altitude (0.205) > soil clay content (0.172) > precipitation (0.163) > soil send content (0.138) > slope (0.059) > aspect (0.014) (Figure 7). These results show that land use, temperature and altitude were the dominant factors in the FVC in this basin. The soil clay content, precipitation and the soil sand content also had key roles, whereas the slope and aspect had the least impact on the FVC. From 2000 to 2020, the *Q* values of all the factors exceeded the average *Q* values in nearly half of the years and the *Q* values of the land use types were always higher than the average explanatory power. The performance of the soil characteristics, topographic conditions and meteorological elements was concentrated before 2010 and gradually began to show higher than average *Q* values after 2017. This suggests that the impact of human activities (change in land use) on the FVC of the basin was the most important factor and the interpretation power for the FVC was the consistently the strongest. The natural factors were affected by other unknown factors from 2010 to 2017 and gradually increased after 2017.

Figure 8 shows that the explanatory power of the interaction of vegetation coverage factors in the Urumqi River basin from 2000 to 2020 formed 28 pairs of interactions after the superposition of the eight factors in the two spaces. Among them, the *Q* values of 16–27 interaction pairs were more significant than the maximum *Q* values of the two impact factors, indicating that their interaction effects on the basin FVC were enhanced in a bivariate manner. The *Q* values of 1–11 interaction pairs were greater than the sum of the two impact factors, indicating that their interaction effects on the basin FVC were enhanced in a nonlinear manner. However, the *Q* value of 1–2 for the interaction was between the *Q* values of the two impact factors, indicating that their interaction effect on the basin FVC was a single-factor, nonlinear, weakening effect. The *Q* value of the interaction between land use and the soil clay content, sand content, altitude, temperature and precipitation was the largest (0.415–0.576), indicating that these interactions had a dominant role in the FVC after spatial superposition, followed by the interaction between other factors outside the slope and aspect (0.17–0.42). The *Q* values of the interaction between slope, aspect and other factors were generally <0.2. In terms of time, the interaction between the factors showed a trend of increasing from 2000 to 2011, decreasing from 2011 to 2017, and slowly recovering after 2017. From the perspective of factors, the soil sand content, slope, altitude, land use, precipitation and temperature all had an auxiliary role in a bivariate enhancement. The slope, soil clay content and soil sand content had a positive, nonlinear strengthening role, whereas the slope aspect and other factors had a nonlinear, weakening role, limiting the FVC of the basin.

#### 3.4.2. Risk Detector

The OPGD package can compute categorical variables directly, but the continuous variables need to be optimally discretized for subsequent operations. To optimize the spatial discretization parameters of the continuous variables, we selected three to nine discontinuous points and four methods (standard deviation, equality, quantile and geometry) and used the letters A–I to denote the nine discontinuous points. Figure 9 shows that the optimum parameter combinations of different explanatory variables are different: the optimum parameter combinations of the continuous variables (altitude, slope and temperature) are equal discontinuities of eight intervals, whereas the optimum parameter combinations of precipitation are the standard deviation of the discontinuities of five intervals. Based on the greatest discretization, we carried out risk detection of the basin year-by-year FVC.

According to the analysis of the soil characteristics risk region based on the OPGD model (Figure 10A), the most suitable area for the FVC of the basin with a surface soil clay content from 2000 to 2022 was >15.38% and the FVC_mean_ was between 0.42 and 0.7. The inhibition region ranged from 0 to 6.67% and the FVC_mean_ ranged from 0.08 to 0.2. In the linear fitting results of the changes in the FVC_mean_ in each interval, we found that the appropriate levels of the FVC_mean_ in the A, D and inhibition regions were higher and showed an upward trend, whereas the changing trend in the other intervals was unclear. Spatially, most of the basin was suitable for the growth of vegetation and the area with a decreased FVC was distributed in the desert area in the lower reaches of the basin (Figure 10C_1) and the area near the glaciers in the upper reaches (Figure 10C_2). The area with an increase in the FVC was concentrated in the buffer zone in the upper and middle reaches of the basin as well as in the suppression area. In general, there was a linear relationship between the clay content of the surface soil and the FVC_mean_. The FVC_mean_ was more extensive in the region with a higher soil clay content (Figure 10B). Only some of the regions showed a significant upward trend from 2000 to 2022, consistent with the spatial distribution of the FVC (Figure 4). By contrast, there was no linear relationship between the sand content of the surface soil and the FVC_mean_ (Figure 10E). The sand content in the area most suitable for FVC ranged from 19.3 to 27% and the average vegetation cover ranged from 0.34 to 0.5. When the sand content was >46.6%, the FVC was inhibited. The average FVC was only 0.05–0.2 (Figure 10D). Spatially, the suitable region was distributed in the downstream, midstream and part of the upstream region. By contrast, the inhibition region was distributed around the suitable region (Figure 10F). In general, an increase in the sand content will decrease the vegetation coverage. The area with a low FVC showed a significant growth trend over time from 2000 to 2020.

Based on the analysis of the terrain feature risk area using the OPGD model (Figure 11A), an altitude between 1300 and 2700 m was suitable for vegetation growth and the FVC_mean_ was between 0.36 and 0.55. An altitude > 3700 m inhibited vegetation growth and the FVC_mean_ was between 0 and 0.17. There was little effect on the FVC when the altitude was <1300 m.

From 2000 to 2020, the suitable area of the FVC_mean_ moved in an orderly manner within the optimum C–E interval by about 450 m every year, indicating that the suitable area had stable and specific characteristics of vertical zonal migration (Figure 11). The elevation showed a significant upward trend in the FVC with time only in intervals G and H. In the suppressed high-elevation region (Figure 11B), the spatially suitable areas were distributed in the mountains in the eastern middle reaches, the buffer zone between the upper and middle reaches, and the upstream riparian zone (Figure 11C). The mid-altitude region was the main region of vegetation coverage in the basin. It was also a relatively stable and self-adjusting region. The clear increase in FVC_mean_ in the high-altitude region may result from its suitability for vegetation caused by the degradation of snow and ice in this region as a result of global climate change.

The aspect affects the duration of sunshine and the intensity of solar radiation. Because the study area is located in the northern hemisphere, the southern slopes receive the most solar radiation, followed by the southeastern and southwestern slopes, then the eastern, western and northeastern slopes. All the slope aspects in the basin were suitable for vegetation, although an area of inhibition was found in flat areas and on the northwestern and northern slopes. In general, the slope of the basin had the greatest effect on the number of hours of sunshine and the amount of solar radiation.

The most suitable slope in the basin was 30–35°, with an FVC_mean_ of 0.2–0.4, whereas the inhibition area had a slope > 55° (Figure 11G). The slope of the suitable area gradually decreased over time. Spatially, the mountains in the east of the middle (Figure 11I_1) and upper (Figure 11I_2) reaches of the basin had complex slopes and an uneven FVC and were found in the between middle and high FVC areas of the basin. In general, the slope of the basin was more complex in the mountainous region, which is an important reason for determining the level of FVC.

According to the analysis of the risk region of climate characteristics based on the OPGD model (Figure 12A), the annual precipitation of the basin ranged from 120 to 575 mm from 2000 to 2020 and the precipitation (170–250 mm) was suitable for the growth of vegetation. The cause of low vegetation cover was precipitation that was either too high or too low. Based on the change in FVC_mean_ in each interval, the relationship between precipitation and vegetation cover was unclear and only FVC levels with high or low precipitation showed a significant change, with a linear growth relationship over time. In general, the precipitation in the basin increased yearly, but the spatial distribution was uneven and the range of the water demand of vegetation was unstable. Precipitation was only suitable for vegetation growth in a specific range (Figure 12B).

From 2000 to 2020, the average annual temperature of the basin ranged from −4 to 31 °C; the temperature range suitable for vegetation growth is 8–16 °C. Vegetation growth was inhibited when the average annual temperature was <0 °C (Figure 12C). In terms of time, the change in the interval outside the extreme temperature range was insignificant, whereas both the A interval of the low-temperature level and the H interval of the high-temperature level, which both inhibited vegetation growth, showed a significantly increasing trend (Figure 12D). In general, the suitability of high-altitude snow- and ice-covered areas was increased by the interaction of multiple factors. The increase in vegetation cover in the desert area with high temperatures is the main reason for the change in vegetation cover with changes in temperature.

According to the regional analysis of land use type risk based on the OPGD model (Figure 13A), farmland and forest were the main contributors to the basin FVC and the areas with the lowest FVC level were the areas of water. In terms of time, the FVC of the basin changed significantly and increased in the order construction land > grass > unused land > water, whereas the farmland and forest did not show any significant change (Figure 13B). In general, the primary trend from 2000 to 2020 was an increase in the vegetation cover in the basin, which confirmed the results of the analysis of the change in areas of construction, grass and unused land. The FVC of the water area was relatively low, but showed a significant upward trend in the last 20 years, exceeding the level of the previous years after 2017.

## 4. Discussion

### 4.1. Significant Spatiotemporal Variation in Vegetation Cover of the Urumqi River Basin

The mean value of the FVC in most areas of the Urumqi River basin showed a fluctuating upward trend from 2000 to 2020, with a rate of increase of 0.0018 (Figure 2). The vegetation cover increased as long as the area of No__FVC_ decreased and Low__FVC_ increased, consistent with the findings of Yaxiao et al. [48] for the vegetation cover on the northern slopes of the Tianshan Mountains. The No__FVC_ and Low__FVC_ areas were mainly distributed in the upper part of the Urumqi River basin near Glacier No. 1, around the built-up areas and in the desert areas with a poor natural environment downstream (Figure 1). However, our results differ from those of Liu et al. for the vegetation cover of the whole territory; in their study, the vegetation cover of the desert areas in Xinjiang showed a decreasing trend and the oasis areas showed a significant increasing trend [49,50,51,52,53]. By contrast, our results showed a significant increase in vegetation cover in both urban and desert areas (Figure 5). The increase in vegetation cover in built-up areas was mainly due to the strengthening of urban greening policies, whereas the increase in vegetation cover in the downstream desert areas was mainly due to the increasing trend of precipitation in the study area over the last 21 years and the successive increases in precipitation in 2008–2014 and 2016–2020 (Figure 12A).

### 4.2. Each Factor Had a Significant Effect on the Vegetation Cover of the Urumqi River Basin

The average strength of the explanatory power of natural factors on vegetation cover over the 21-year study period varied with temperature (0.216) > elevation (0.205) > soil clay content (0.172) > precipitation (0.163) > soil sand content (0.138) > slope (0.059) > slope orientation (0.014) (Figure 6). Changes in vegetation cover are complex processes and are significantly influenced by human activities (land use type, GDP density, population density and distance from roads) [23,54,55]. Previous studies in Xinjiang have shown that human activities and active policies are the main influences on the regional vegetation cover [56]. Land use types, as a map of human activities, have a dominant role in the distribution of vegetation cover, of which the most discussed types are arable land, water bodies and construction land. The variation in area between low and medium vegetation cover in croplands indicates that the cropping pattern in the basin is mainly one of shifting cultivation. This result is closely related to the way in which croplands are managed in arid areas [57,58]. When analyzing the vegetation cover of water bodies, we found that, in addition to the classification accuracy of land use types, Low_FVC and Medium_FVC water bodies are caused by the growth of aquatic plants [59]. The vegetation cover of construction land showed a continuously increasing trend. Human activities have therefore had a positive impact on the growth of vegetation in the Urumqi River basin. This conclusion was confirmed in a study by Zhang et al. [47] on the relationship between global urbanization and vegetation growth.

Previous studies of the influence of natural factors on vegetation cover have shown that temperature has the greatest effect on vegetation cover in the high-altitude areas of the Tibetan Plateau and Sichuan Province, whereas precipitation has a more significant effect on changes in vegetation cover in most semi-arid regions [60,61,62]. Our study showed that both temperature and precipitation have large effects in the Urumqi River basin, which could be due to its special geographical environment of a typical arid zone with high mountains and deserts coexisting in the basin [63]. We found that the regions with an increasing or decreasing FVC were relatively complex and the overall FVC increased slowly over time. However, the influence of soil characteristics and topographic conditions in the basin was spatially unique. The clay and sand contents of the surface soils were all suitable for medium levels of FVC. At altitudes of 1300–2700 m, the southern, southeastern and southwestern slopes, with higher levels of solar radiation and slope aspects suitable for water transport, were all areas suitable for the growth of vegetation. The desert areas showed a low-level upward trend, whereas the mountainous areas showed a downward trend. In the high-altitude areas, the interaction of multiple factors showed a unique distribution of vegetation [64,65,66].

From 2000 to 2020, in addition to relatively stable soil characteristics and topographic conditions, the annual precipitation, annual average temperature and human activities showed overall growth and differences in distribution in the Urumqi River basin. The comprehensive effect of these factors is an important reason for the change in vegetation cover in this basin. Previous studies of the influence of climate factors on the FVC have shown that precipitation is the primary influence on the FVC and that significant changes occur around glaciers, snow cover and lakes in mountainous regions [67,68,69,70]. We found that the higher the annual precipitation, the higher the vegetation coverage in the Urumqi River basin. However, the areas with the highest vegetation cover were distributed in areas with an average annual precipitation, indicating that there were other factors contributing to the effect of precipitation. The average annual temperature is an important climatic factor for vegetation growth. In the temperature range most suitable for vegetation growth (8–16 °C), the vegetation is adapted to the conditions of the alpine and high drought areas and there is a clear regional growth trend after the melting of snow and ice. In previous studies, human activities led to environmental degradation and the loss of both soil and water.

### 4.3. Shortcomings and Research Significance

Arid regions are more vulnerable to natural and human activities than other regions [6,71]. Urumqi City, the economic, political and cultural center of Xinjiang, is located in the Urumqi River basin. With the construction of the Belt and Road Initiative in recent decades, Urumqi City has experienced rapid urbanization as the core driving region for the Silk Road Economic Belt [72,73]. Global warming and the melting of glaciers have significantly affected the landscape pattern of the basin, resulting in significant changes in the vegetation cover [74,75]. Previous studies used a geographical detector model approach to analyze the effects of various factors on vegetation cover, focusing on single years and subjective discretization [54,76]. Under this background, we used the OPGD model to monitor the Urumqi River basin over a long time period at high spatial resolution in an attempt to understand the intrinsic mechanisms of these changes. However, there were some limitations to our study—for example, we used remote sensing images at a certain time in the growth season of vegetation to represent one year of data. The influencing factors selected did not consider all the natural (e.g., soil fertility and soil moisture) and human (e.g., road distribution and population) factors.

Despite these limitations, we effectively quantified the relative contributions of the main drivers of changes in vegetation cover and their interactions over time, as well as identifying the areas suitable and inhibited for vegetation cover. The findings of this study on the vertical zonal migration of vegetation have important implications for the management and construction of ecological vegetation for upstream water conservation. The results of the study will be helpful in understanding the changes in vegetation cover and its driving factors in small-scale basins in arid regions and will provide a reference for ecological conservation in Xinjiang.

## 5. Conclusions

The characteristics of climate, topography and human activities in the Urumqi River basin are typical of arid and semi-arid areas. We analyzed the trends and driving factors of vegetation coverage in the Urumqi River basin from 2000 to 2020 and drew the following conclusions.

From 2000 to 2020, the FVC_mean_ of the Urumqi River basin was 0.25–0.30, the decrease in the mean value varied regularly with time and the overall growth was stable. The vegetation coverage was mainly low and medium, and the areas without vegetation coverage decreased steadily year-by-year, whereas the areas with medium and high vegetation coverage maintained an area of 600–1000 km^2^.The FVC of the basin had unique spatial characteristics. The areas without vegetation were distributed at high altitudes, on the urban fringes and in areas with an extreme environment. The areas without vegetation all improved, except the high-altitude areas. Areas with low vegetation coverage formed around each type of coverage area and became a critical ecological barrier, making vital contributions to the basin. The areas with medium vegetation coverage were mainly distributed in the mountains and plains and were stable overall.The vegetation coverage of the basin was affected by both natural and human factors. The explanatory power of land use type, precipitation, altitude and temperature was the highest and the explanatory power of interaction with other factors was increased and had the greatest impact. The explanatory power of other factors was small and the interaction results were not significantly improved. As the only anthropogenic factor, the explanatory power of the land use type was maintained at a high level and indicated that human habitation and activities significantly affected the trends of the changes in vegetation cover. The explanatory power of each factor interval differed. The altitude, slope and clay content of surface soils had adjacent values for their effect on vegetation and stopped vegetation from growing when out of its natural range. Precipitation and temperature did not show a positive correlation, but had a promoting effect locally. The growth of vegetation under drought conditions in cold, high-altitude regions actively adapted to the local environment, which may be related to the unique climate characteristics of the study area and vegetation types.As the open-source multispectral remote sensing data archive enriches (i.e., Landsat 9 and Sentinel 2), we will be able to extend the temporal coverage of the FVC. In addition, UAV technology enables exploration FVC of small-scale areas, and vegetation health can be evaluated by the red-edge bands of, for example, a Sentinel 2 image. Thus, future attempts would backdate the FVC monitoring to monthly, while analyzing the relationship between human habitat, water pollution, air pollution and FVC.

## Figures and Tables

**Figure 1 ijerph-19-15323-f001:**
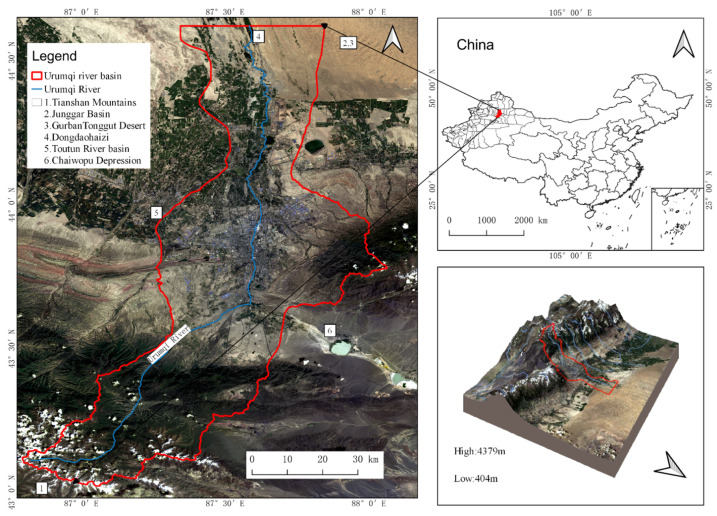
Location of the Urumqi River basin.

**Figure 2 ijerph-19-15323-f002:**
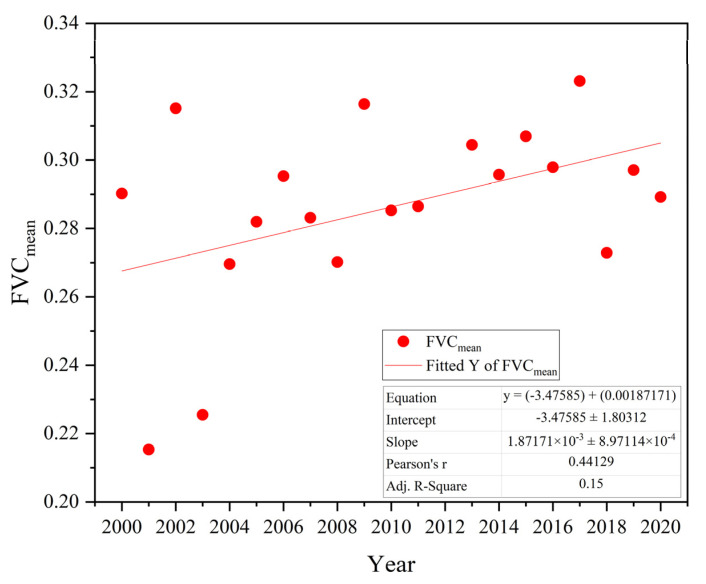
Changes and trends of FVC_mean_ from 2000 to 2020.

**Figure 3 ijerph-19-15323-f003:**
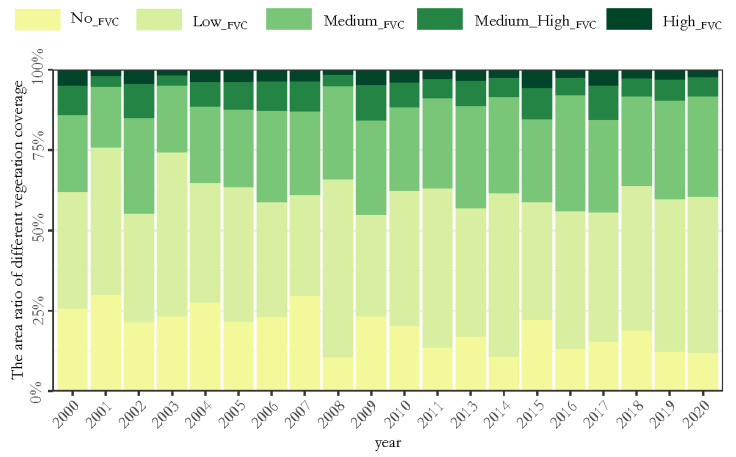
Annual changes in areas with a different FVC from 2000 to 2020.

**Figure 4 ijerph-19-15323-f004:**
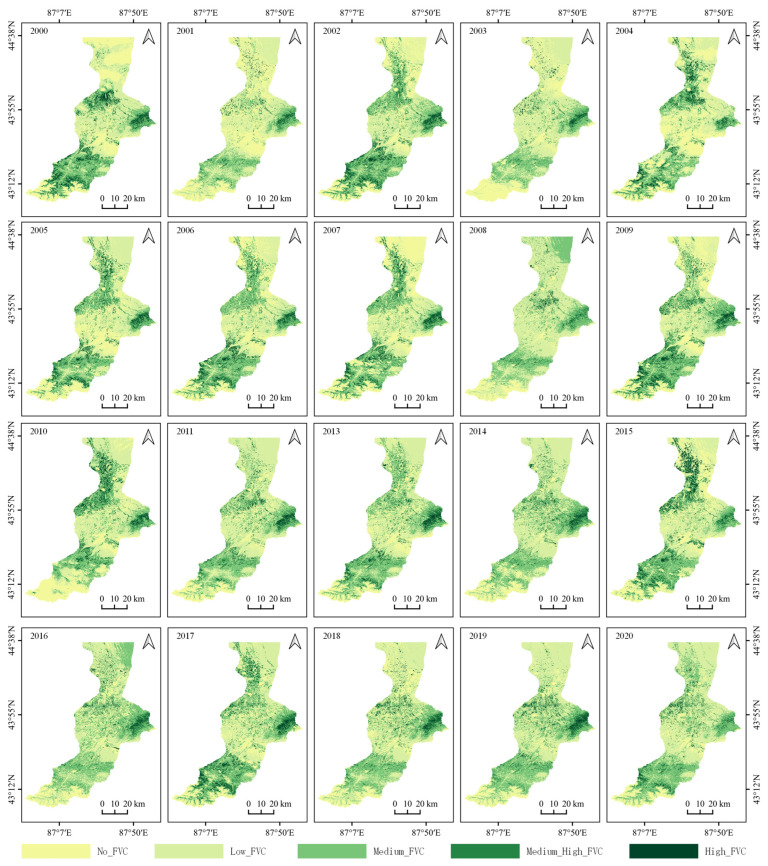
Annual changes in the distribution of vegetation coverage from 2000 to 2020.

**Figure 5 ijerph-19-15323-f005:**
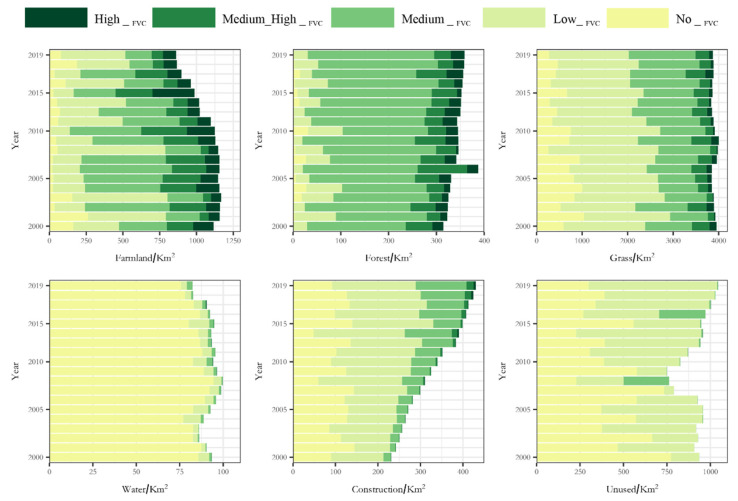
Proportion of FVC for each land use type from 2000 to 2019.

**Figure 6 ijerph-19-15323-f006:**
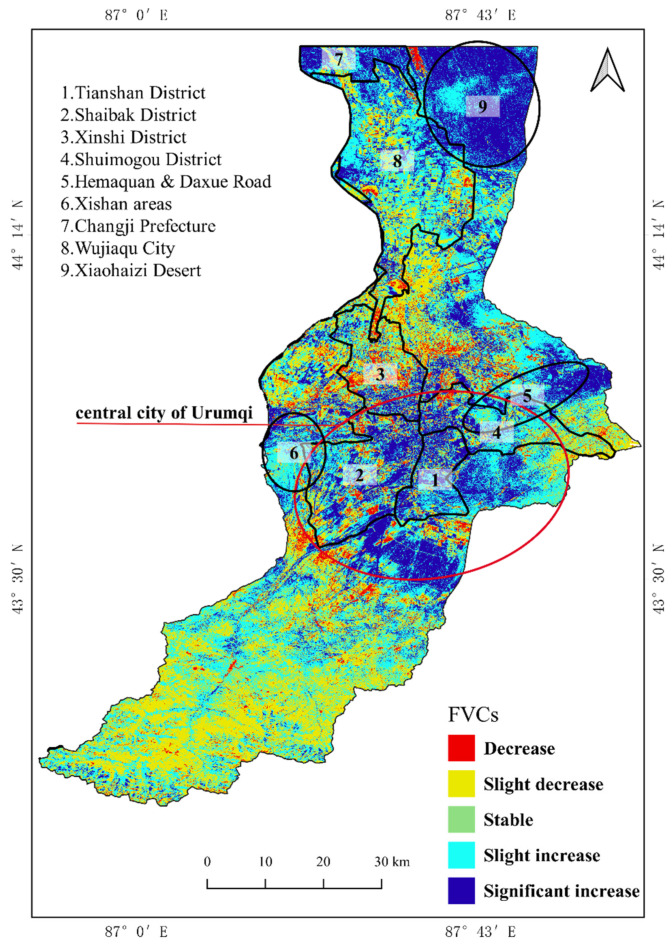
Trend of the FVC from 2000 to 2020.

**Figure 7 ijerph-19-15323-f007:**
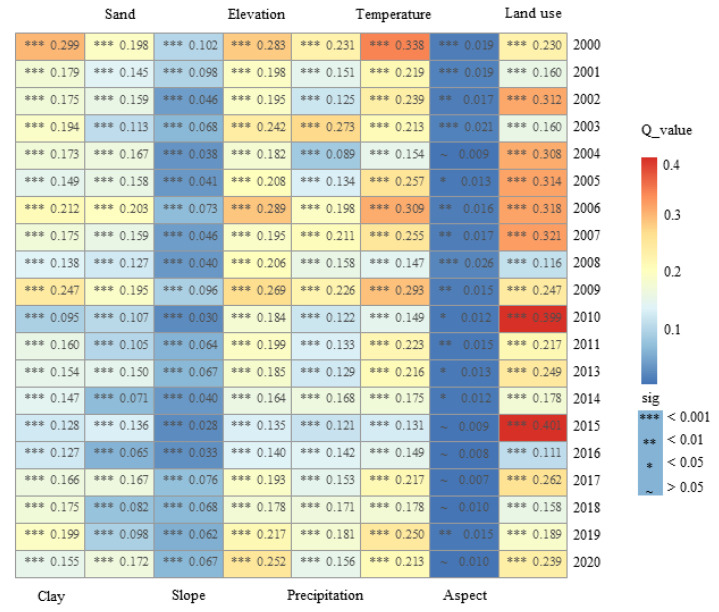
*Q* values of the FVC factor detector from 2000 to 2020.

**Figure 8 ijerph-19-15323-f008:**
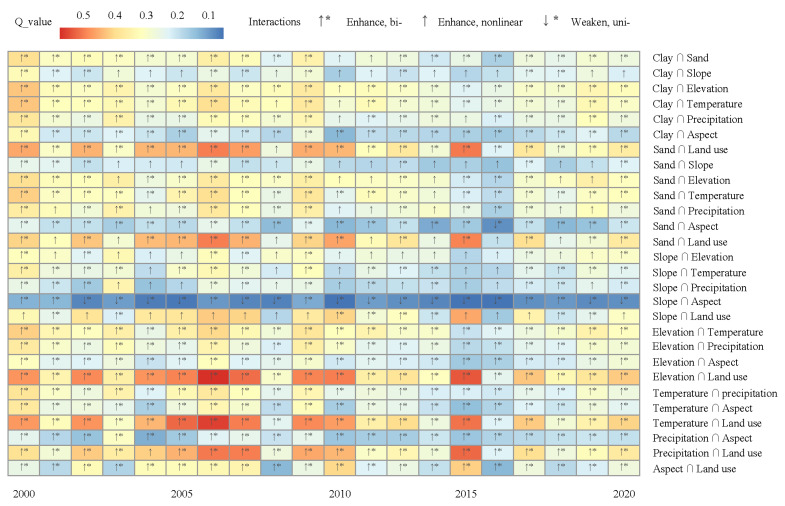
*Q* values of the FVC interaction detector from 2000 to 2020.

**Figure 9 ijerph-19-15323-f009:**
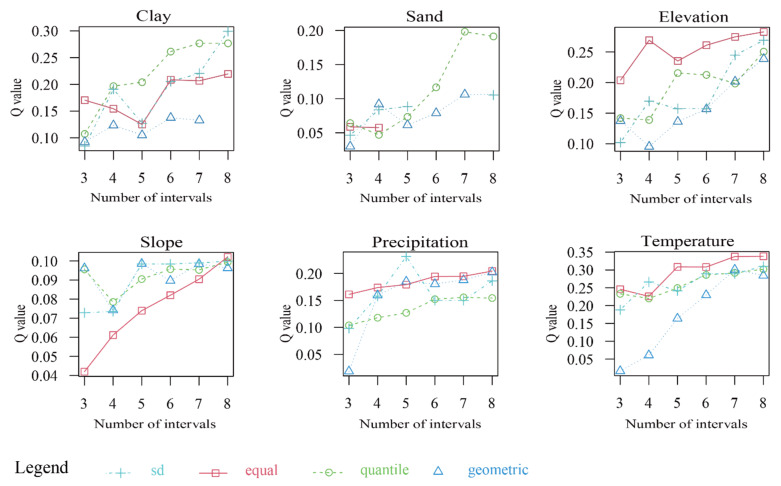
Parameter optimization process for the discretization of spatial data.

**Figure 10 ijerph-19-15323-f010:**
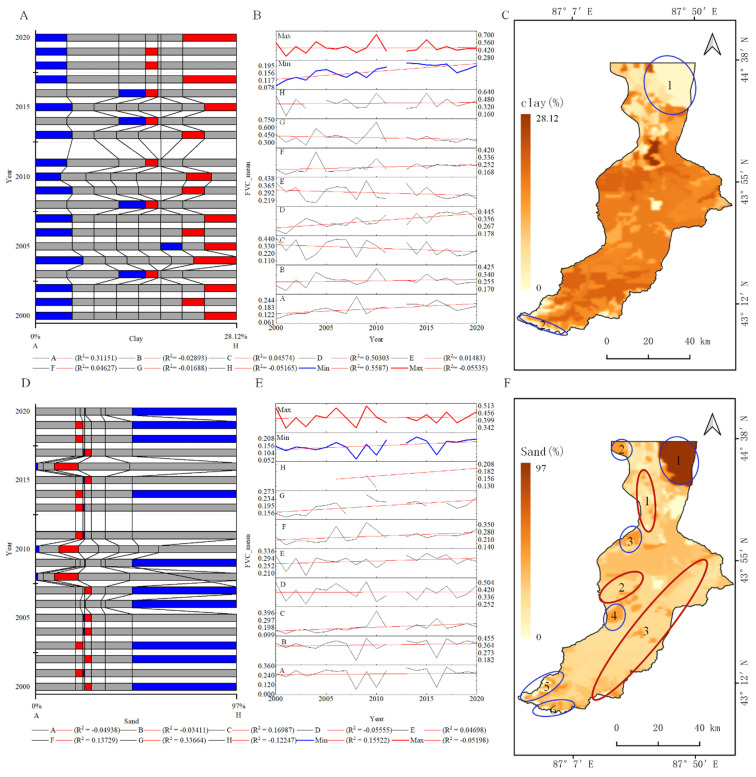
Analysis of the soil characteristic risk area based on the OPGD model: (**A**) soil clay content risk areas; (**B**) FVC_mean_ values for each soil clay content risk areas; (**C**) spatial distribution of soil clay content; (**D**) soil sand content risk areas; (**E**) FVC_mean_ values for each soil sand content risk areas; (**F**) spatial distribution of soil sand content; (red line) maximum FVC_mean_ value in risk area; (blue line) minimum FVC_mean_ value in risk area.

**Figure 11 ijerph-19-15323-f011:**
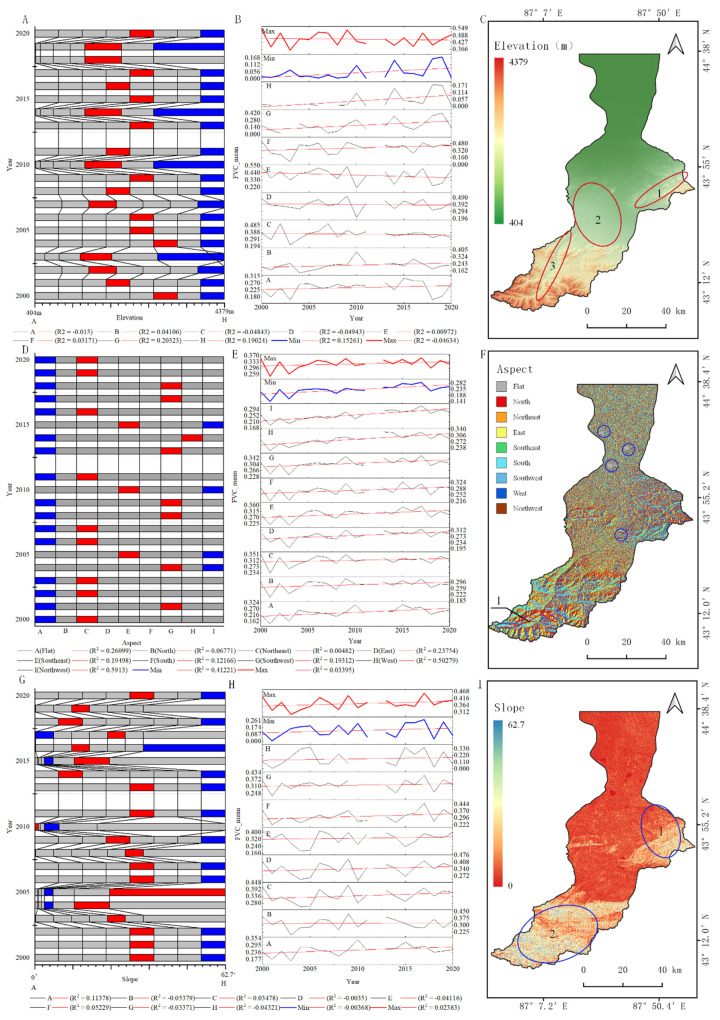
Analysis of the terrain characteristic risk area based on the OPGD model: (**A**) elevation risk areas; (**B**) FVC_mean_ values for each elevation risk areas; (**C**) spatial distribution of elevation; (**D**) aspect risk areas; (**E**) FVC_mean_ values for each aspect risk areas; (**F**) spatial distribution of aspect; (**G**) slope risk areas; (**H**) FVC_mean_ values for each slope risk areas; (**I**) spatial distribution of slope; (red line) maximum FVC_mean_ value in risk area; (blue line) minimum FVC_mean_ value in risk area.

**Figure 12 ijerph-19-15323-f012:**
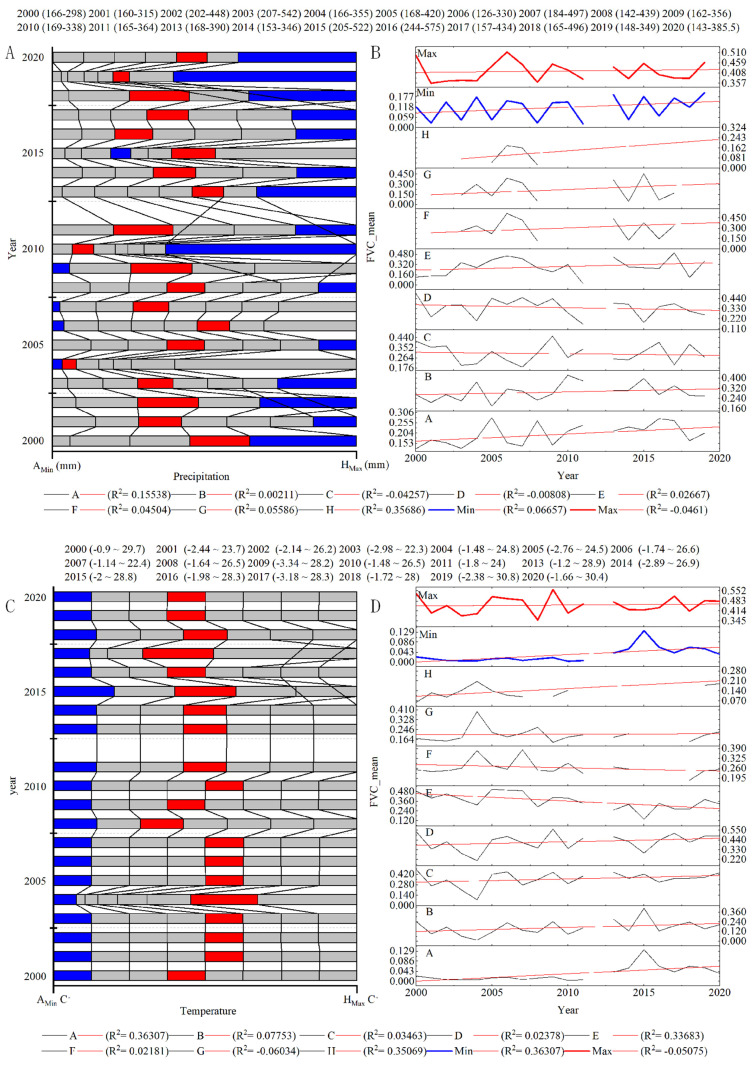
Analysis of the climate characteristic risk area based on the OPGD model: (**A**) precipitation risk areas; (**B**) FVC_mean_ values for each precipitation risk areas; (**C**) temperature risk areas; (**D**) FVC_mean_ values for each temperature risk areas; (red line) maximum FVC_mean_ value in risk area; (blue line) minimum FVC_mean_ value in risk area.

**Figure 13 ijerph-19-15323-f013:**
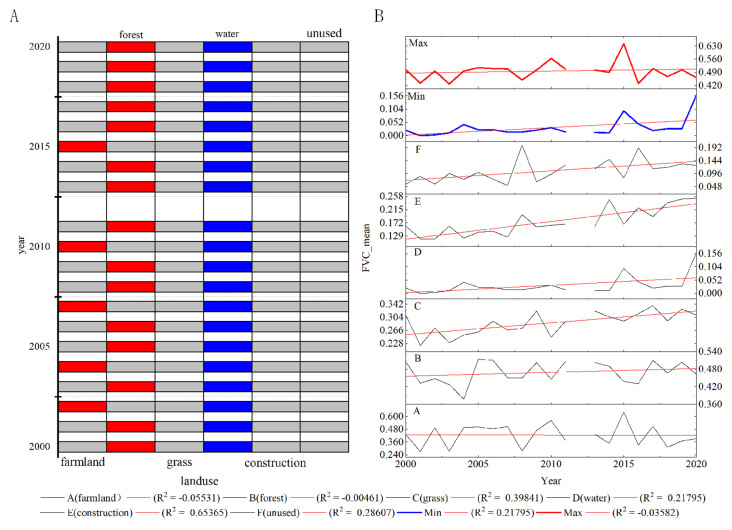
Analysis of the land use risk area based on the OPGD model: (**A**) land use type risk areas; (**B**) FVC_mean_ values for each land use type risk areas; (red line) maximum FVC_mean_ value in risk area; (blue line) minimum FVC_mean_ value in risk area.

**Table 1 ijerph-19-15323-t001:** Trend statistics of the FVC.

Significance Level	*Z* Value	Trend of FVC	Area (km^2^)	Percentage (%)
≥0.0005	≥1.96	Significant increase	1738.59	26.15
≥0.0005	−1.96 to 1.96	Slight increase	2419.39	36.39
−0.0005 to 0.0005	−1.96 to 1.96	Stable	376.31	5.66
Less than −0.0005	−1.96 to 1.96	Slight decrease	1790.44	26.93
Less than −0.0005	Less than −1.96	Decrease	323.78	4.87

## Data Availability

Not applicable.

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
