# Peer review of "Spatiotemporal Evolution and Driving Forces of Vegetation Cover in the Urumqi River Basin"

_ijerph, 2022, doi:10.3390/ijerph192215323_

Round 1
Reviewer 1 Report
This manuscript presents a way of evaluating the quality of regional ecology in the basin by the fractional vegetation coverage (FVC) and the optimal parameters-based geographical detector (OPGD) model to quantitatively analyze the influence of the factors on the change in vegetation coverage from 2000 to 2020.
This package can be applicable on a large scale in several situations and regions around the world as an essential indicator of the quality of the regional ecological environment.
The methodology is very well written, with clear and direct language, presenting enough details for the reproduction of the research.
The innovative aspect of this research is the use of medium- and high-resolution remote sensing images to study the river basin in typical arid and semi-arid areas by the geographical detector is a statistical method that reveals the influencing factors such as climate, vegetation, topography, terrain, and soils.
These results show that land use, temperature, and altitude were the dominant factors in the FVC in this watershed in China. The impact of human activities (change in land use) on the FVC was the most important factor and the interpretation power for the FVC was consistently the strongest.
All tables, figures, and equations are properly cited in the text, and all citations are listed in the references section.
The manuscript is suitable for publication after minor revisions and adjustments pointed out in the revised file (see attached system).

Author Response
List of Responses
Dear Editors and Reviewers:
Thank you for your letter and for the reviewers’ comments concerning our manuscript entitled “Spatiotemporal evolution and driving forces of vegetation cover in the Urumqi River basin” (Manuscript ID: ijerph-1949742). Those comments are all valuable and very helpful for revising and improving our paper, as well as the important guiding significance to our research. We have studied the comments carefully and have made corrections which we hope meet with approval. The main corrections in the paper and the responses to the reviewer’s comments are as flowing:
Responds to the reviewer’s comments:
- Response to comment: (The manuscript is suitable for publication after minor revisions and adjustments pointed out in the revised file (see attached system))
Response: We have made corrections according to the Reviewer’s comments.
Special thanks to you for your good comments.

Reviewer 2 Report
1. Figure 1 shows the study area in a way that is not very intuitive and could be presented in the form of a zoomed map. The two administrative district vector maps on the right are not obvious enough the highlighting the study area. In particular, the legend of numbers 1-5 on the left map is not accurate
2. The Data section needs to be streamlined, it looks too cumbersome and can be summarized in tabular form
3. Was the NDVI used averaged from June to August or was the image just selected for one of those times?
4. Was the NDVIsoil used in the calculation of FVC directly taken as zero or are ground sample points selected for the statistics? Are the NDVIsoil and NDVIveg selected for each year robust and comparable?
5. Why did the Northeast region appear so much Medium_FVC in 2008 and 2016? This region should be desert dominated. Please provide the original image and the corresponding NDVI image to support it
6. what's the meaning of the watershed area? Where is the watershed area? The expressions of watershed and basin should be unified to avoid ambiguity
7. In 3.2, “FVC accounted for 94.34% of the total watershed area (about 6640.8855 km2)” and“The area with an increasing trend of vegetation coverage accounted for 62.56% of the total watershed area (about 4158.1702 km2)”. The total area in these two sentences is not the same
8. What is the meaning of the leftmost column in Table 1? Is it the range of probability values to reject the hypothesis? The alpha of significant change should be less than or equal to 0.05 confidence level
9. Why does the regional vegetation cover in Urumqi increase on built-up land? Did urban greening or cultivated land reclamation cause it? Or is it because there is a problem with the calculation formula, such as the NDVIsoil calibration?
10. The factors analyzed by OPGD are all at 1000m resolution, but the FCV is at 30m. Is the final spatial resolution of the analysis at 1000m or 30m?
11. What do the various color blocks in Figures A and D in Figure 9 mean? What do the line graphs A-H in Figures 9 through 12 indicate? The legend A-I in Figure 10. F should be changed to slope direction. Please note the accuracy and clarity of the presentation of all graphs!
12. Does FVCmean indicate the average FVC for the year? Is it a regional average or a pixel average? Please clarify it.
13. Is the land use data taken from year to year? Why does land use as cropland have no effect on FVC? Moreover, the FVC should be zero on construction land and unused land, so why is there such a large increase in FVC on these land use? What causes the increase of FVC in the construction land and unused land?
14. It is unclear how the significant and non-significant changes in vegetation cover and the trends of increase and decrease are related to each factor in temporal and spatial terms。
15. The discussion is too superficial and does not go into the causal relationship between FVC changes and regional climatic and topographic features. It could be useful to consider the implications of the results of the OPGD analysis for regional ecosystem management.
Author Response
List of Responses
Dear Editors and Reviewers:
Thank you for your letter and for the reviewers’ comments concerning our manuscript entitled “Spatiotemporal evolution and driving forces of vegetation cover in the Urumqi River basin” (Manuscript ID: ijerph-1949742). Those comments are all valuable and very helpful for revising and improving our paper, as well as the important guiding significance to our research. We have studied the comments carefully and have made corrections which we hope meet with approval. The main corrections in the paper and the responses to the reviewer’s comments are as flowing:
Responds to the reviewer’s comments:
- Response to comment: (Figure 1 shows the study area in a way that is not very intuitive and could be presented in the form of a zoomed map. The two administrative district vector maps on the right are not obvious enough the highlighting the study area. In particular, the legend of numbers 1-5 on the left map is not accurate)
Response: We have made corrections according to the Reviewer’s comments.
- Response to comment: (The Data section needs to be streamlined, it looks too cumbersome and can be summarized in tabular form)
Response: Because of the large number of data presentation, it is difficult to layout after making a table. I hope you will allow me to keep such expressions.
- Response to comment: (Was the NDVI used averaged from June to August or was the image just selected for one of those times?)
Response: We have made corrections according to line 126 to the Reviewer’s comments.
- Response to comment: (Was the NDVIsoil used in the calculation of FVC directly taken as zero or are ground sample points selected for the statistics? Are the NDVIsoil and NDVIveg selected for each year robust and comparable?)
Response: The unvegetated bare land (NDVIsoil) and high vegetation cover areas (NDVIveg) were considered in addition to the distribution of image elements at 5% and 95%, and fieldwork experience was fully considered. Because it is a long time series data, it is difficult to determine a point to complete.
- Response to comment: (Why did the Northeast region appear so much Medium_FVC in 2008 and 2016? This region should be desert dominated. Please provide the original image and the corresponding NDVI image to support it)
Response: The precipitation in the watershed was higher in all these years, also mentioned in the conclusion and discussion in the paper. Desert areas have more desert vegetation such as pokeweed, but such vegetation is difficult to see and cast a color with the subsurface in the original images. Therefore, I cannot provide a clearer comparison image. But I will use high resolution remote sensing images to improve this work in the future.
- Response to comment: (what's the meaning of the watershed area? Where is the watershed area? The expressions of watershed and basin should be unified to avoid ambiguity)
Response: We have made corrections according to the Reviewer’s comments.
- Response to comment: (In 3.2, “FVC accounted for 94.34% of the total watershed area (about 6640.8855 km2)” and “The area with an increasing trend of vegetation coverage accounted for 62.56% of the total watershed area (about 4158.1702 km2)”. The total area in these two sentences is not the same.)
Response: We have made corrections according to the Reviewer’s comments.
- Response to comment: (What is the meaning of the leftmost column in Table 1? Is it the range of probability values to reject the hypothesis? The alpha of significant change should be less than or equal to 0.05 confidence level)
Response: We have made corrections according to the Reviewer’s comments.
- Response to comment: (Why does the regional vegetation cover in Urumqi increase on built-up land? Did urban greening or cultivated land reclamation cause it? Or is it because there is a problem with the calculation formula, such as the NDVIsoil calibration?)
Response: We have made corrections according in the discussion to the Reviewer’s comments.
- Response to comment: (The factors analyzed by OPGD are all at 1000m resolution, but the FCV is at 30m. Is the final spatial resolution of the analysis at 1000m or 30m?)
Response: We have made corrections according to lines 183_184 to the Reviewer’s comments.
- Response to comment: (What do the various color blocks in Figures A and D in Figure 9 mean? What do the line graphs A-H in Figures 9 through 12 indicate? The legend A-I in Figure 10. F should be changed to slope direction. Please note the accuracy and clarity of the presentation of all graphs!)
Response: We have made corrections according to the Reviewer’s comments.
- Response to comment: (Does FVCmean indicate the average FVC for the year? Is it a regional average or a pixel average? Please clarify it.)
Response: We have made corrections according to line 257 to the Reviewer’s comments.
- Response to comment: (Is the land use data taken from year to year? Why does land use as cropland have no effect on FVC? Moreover, the FVC should be zero on construction land and unused land, so why is there such a large increase in FVC on these land use? What causes the increase of FVC in the construction land and unused land?)
Response: We have made corrections according to lines 300_333 to the Reviewer’s comments.
- Response to comment: (It is unclear how the significant and non-significant changes in vegetation cover and the trends of increase and decrease are related to each factor in temporal and spatial terms.)
Response: This work I completed initially. But the explanatory power of each year driver to calculate the trend of change is inaccurate. Therefore, the trend of each factor was used to explain the trend of vegetation cover change, and good results were obtained. However, this result is difficult to illustrate the problem and there is no such precedent. I will provide these data in the attachment, but was not able to organize them particularly well within this event due to my own pre-infection with the novel coronavirus.
- Response to comment: (The discussion is too superficial and does not go into the causal relationship between FVC changes and regional climatic and topographic features. It could be useful to consider the implications of the results of the OPGD analysis for regional ecosystem management.)
Response: We have made corrections according in the discussion to the Reviewer’s comments.
Special thanks to you for your good comments.

Reviewer 3 Report
Personal voice should be changed into impersonal passive voice.
The introduction does not formulate a clear and substantive research goal. Admittedly, it mentions the conducted research, but what was the research objective and research questions? They should be strongly emphasized in both the introduction and the abstract.
The conducted research very briefly touches upon vegetation–which is of key importance for the studied topic. The flora of the research area should be discussed more broadly by characterizing the dominant classes and plant associations.
Figure 1 – the location of the region in the country is missing (or illegible). It should be either added to the map or corrected. In order to improve legibility, red marking is proposed for the Xinjiang region, as well as for the studied area. In addition, the surface model should be much larger as it is more important than location maps, and its readability is the lowest, which is a significant shortcoming.
Figure 2 – the readability of the table should be improved, as well as in Figure 8 of the legend. The year 2012 is missing in Figures 3 and 4. What is more, the numbers on the map are poorly visible in Figure 5.
The article contains numerous figures (as many as 11), therefore a slight reduction in the number of graphics should be considered.
It is suggested to use the website references (lines 128; 152; 165) as source material. The literature does not include sources that were only mentioned in the text and not quoted.
In the "Method" subchapter, the legibility of the formulas should be improved, as currently they blend in with the text. More spacing should be applied. The formulas could also be centered.
Furthermore, there is no spacing before the next subsection, which also reduces clarity of the text.
Author Response
List of Responses
Dear Editors and Reviewers:
Thank you for your letter and for the reviewers’ comments concerning our manuscript entitled “Spatiotemporal evolution and driving forces of vegetation cover in the Urumqi River basin” (Manuscript ID: ijerph-1949742). Those comments are all valuable and very helpful for revising and improving our paper, as well as the important guiding significance to our research. We have studied the comments carefully and have made corrections which we hope meet with approval. The main corrections in the paper and the responses to the reviewer’s comments are as flowing:
Responds to the reviewer’s comments:
- Response to comment: (The introduction does not formulate a clear and substantive research goal. Admittedly, it mentions the conducted research, but what was the research objective and research questions? They should be strongly emphasized in both the introduction and the abstract.)
Response: We have made corrections according to lines 1_2 to the Reviewer’s comments.
- Response to comment: (The conducted research very briefly touches upon vegetation–which is of key importance for the studied topic. The flora of the research area should be discussed more broadly by characterizing the dominant classes and plant associations.)
Response: The point you made is indeed a drawback of the article. However, the article proceeds on the premise that all vegetation is one, and after adding the vegetation type of the study area it is difficult to analyze it in place with my knowledge in this area and for the reason of the article content. However, this point will be fully considered in the later study.
- Response to comment: (Figure 1 – the location of the region in the country is missing (or illegible). It should be either added to the map or corrected. In order to improve legibility, red marking is proposed for the Xinjiang region, as well as for the studied area. In addition, the surface model should be much larger as it is more important than location maps, and its readability is the lowest, which is a significant shortcoming.)
Response: We have made corrections according to the Reviewer’s comments.
- Response to comment: (Figure 2 – the readability of the table should be improved, as well as in Figure 8 of the legend. The year 2012 is missing in Figures 3 and 4. What is more, the numbers on the map are poorly visible in Figure 5)
Response: We have made corrections according to the Reviewer’s comments.
- Response to comment: (The article contains numerous figures (as many as 11), therefore a slight reduction in the number of graphics should be considered.)
Response: These icons are really important in the article. Yes, there was a thought to reduce it, but nothing could be done about it. There is also a part put into the appendix. I hope it will be understood.
- Response to comment: (It is suggested to use the website references (lines 128; 152; 165) as source material. The literature does not include sources that were only mentioned in the text and not quoted.)
Response: We have made corrections according to the Reviewer’s comments.
- Response to comment: (In the "Method" subchapter, the legibility of the formulas should be improved, as currently they blend in with the text. More spacing should be applied. The formulas could also be centered)
Response: We have made corrections according to the Reviewer’s comments.
- Response to comment: (Furthermore, there is no spacing before the next subsection, which also reduces clarity of the text.)
Response: We have made corrections according to the Reviewer’s comments.
Special thanks to you for your good comments.

Reviewer 4 Report
General Comments:
Interesting manuscript regarding trends in vegetation cover in the Urumqi River basin and what is contributing to the variation in these trends. The manuscripts contain a lot of data, but the methodology is unclear, presentation partly lacking and discussion in need of expansion. See below for further details.
It is unclear if any quality assessment has been conducted on the Landsat images used for the analysis. It is unclear when these images were taken, and how many images constitute one year or season within the analysis. It is unclear what the overall resolution was of the final assessment, as each dataset have different resolution, leading to the uncertainty of how each product was down/upscaled and how they were all finally combined.
It is clear from reading the manuscript that land use had an important part in the FVC, but the details of how land use changed over the period and which land use that caused more changes are mostly glossed over. The manuscript could use with a figure such as Figure 3, but with land use since it would help the reader to understand the overall trends. Additionally, changes from land use are expected to occur long-term and unidirectionally, then why is there a big difference in the effect of land use between years? This could need a more in-depth explanation. Lastly, changes in vegetation cover might remain stable as natural vegetation is replaced by agriculture or production forest, but the turn-over would be high and cause biodiversity loss for example, this is not talked about either and should be included to some extent.
The discussion section mostly reiterates the result section, without bringing in new information about comparisons to other studies or regions, neither does it try to explain how the findings of the study has created a new understanding of these ecosystems. In relation to the previous comment, the contribution of land use and how the trends in land use are likely to impact the change in FVC are probably the main findings of the study, but it is barely mentioned and must be expanded. Supposedly different land use types should have a varying degree of inherent FVC, with potential thinning and thickening of the inherent FVC and relative changes in different types of land use being important factors in this dynamic.
Specific Comments:
1. Lines 21-22. This sentence might need reformatting, perhaps by adding the word ‘respectively’ to relay that the sentence is bi-folded.
2. Lines 44-46. This sentence does not actually explain what the FVC is. Perhaps add a sentence before this one describing that the FVC is the percentage of living vegetation structures in a given area when compared to non-living structures and viewed from above in a remote-sensing 2D-framework, or similar.
Figures and Tables:
1. Figure 1. The inset (3D topographic map) of this figure would benefit of being its own figure in high resolution, since it would give the reader a good overview of the topography and ecological implications of the area as a whole. Also, Xinjiang is not adequately marked on the overview map.
2. Figure 8. Needs to be made bigger, it is difficult to see any details. Perhaps if it was made from a 6x1 figure to a 3x2 or similar it might make it easier.
3. Figures 9 – 12. These four figures need a much bigger figure text describing each separate part, the figures should be able to be interpreted as stand-alone without the text.
4. Figure 10 C. Is the figure legend containing elevation turned the wrong way here? The red/orange parts should be the higher elevation.
5. Table 1. This table is unclear, the Z-value part of the table needs reformatted since currently it is unclear what is meant by these numbers in relation to the rest of the table.
Author Response
List of Responses
Dear Editors and Reviewers:
Thank you for your letter and for the reviewers’ comments concerning our manuscript entitled “Spatiotemporal evolution and driving forces of vegetation cover in the Urumqi River basin” (Manuscript ID: ijerph-1949742). Those comments are all valuable and very helpful for revising and improving our paper, as well as the important guiding significance to our research. We have studied the comments carefully and have made corrections which we hope meet with approval. Revised portions are marked in yellow on the paper. The main corrections in the paper and the responses to the reviewer’s comments are as flowing:
Responds to the reviewer’s comments:
- Response to comment: (It is unclear if any quality assessment has been conducted on the Landsat images used for the analysis. It is unclear when these images were taken, and how many images constitute one year or season within the analysis. It is unclear what the overall resolution was of the final assessment, as each dataset have different resolution, leading to the uncertainty of how each product was down/upscaled and how they were all finally combined)
Response: We have made corrections according to lines 126, 130, 183_184 to the Reviewer’s comments.
- Response to comment: (It is clear from reading the manuscript that land use had an important part in the FVC, but the details of how land use changed over the period and which land use that caused more changes are mostly glossed over. The manuscript could use with a figure such as Figure 3, but with land use since it would help the reader to understand the overall trends. Additionally, changes from land use are expected to occur long-term and unidirectionally, then why is there a big difference in the effect of land use between years? This could need a more in-depth explanation. Lastly, changes in vegetation cover might remain stable as natural vegetation is replaced by agriculture or production forest, but the turn-over would be high and cause biodiversity loss for example, this is not talked about either and should be included to some extent.)
Response: We have made corrections according to lines 300_333, 552_556 to the Reviewer’s comments.
- Response to comment: (The discussion section mostly reiterates the result section, without bringing in new information about comparisons to other studies or regions, neither does it try to explain how the findings of the study has created a new understanding of these ecosystems. In relation to the previous comment, the contribution of land use and how the trends in land use are likely to impact the change in FVC are probably the main findings of the study, but it is barely mentioned and must be expanded. Supposedly different land use types should have a varying degree of inherent FVC, with potential thinning and thickening of the inherent FVC and relative changes in different types of land use being important factors in this dynamic.)
Response: We have made corrections according to the Reviewer’s comments.
- Response to comment: (Lines 21-22. This sentence might need reformatting, perhaps by adding the word ‘respectively’ to relay that the sentence is bi-folded.)
Response: We have made corrections according to the Reviewer’s comments.
- Response to comment: (Lines 44-46. This sentence does not actually explain what the FVC is. Perhaps add a sentence before this one describing that the FVC is the percentage of living vegetation structures in a given area when compared to non-living structures and viewed from above in a remote-sensing 2D-framework, or similar.)
Response: We have made corrections according to the Reviewer’s comments.
- Response to comment: (Figure 1. The inset (3D topographic map) of this figure would benefit of being its own figure in high resolution, since it would give the reader a good overview of the topography and ecological implications of the area as a whole. Also, Xinjiang is not adequately marked on the overview map.)
Response: We have made corrections according to the Reviewer’s comments.
- Response to comment: (Figure 8. Needs to be made bigger, it is difficult to see any details. Perhaps if it was made from a 6x1 figure to a 3x2 or similar it might make it easier.)
Response: We have made corrections according to the Reviewer’s comments.
- Response to comment: (Figure 10 C. Is the figure legend containing elevation turned the wrong way here? The red/orange parts should be the higher elevation.)
Response: We have made corrections according to the Reviewer’s comments.
- Response to comment: (Table 1. This table is unclear, the Z-value part of the table needs reformatted since currently it is unclear what is meant by these numbers in relation to the rest of the table.)
Response: We have made corrections according to the Reviewer’s comments.
Special thanks to you for your good comments.

Round 2
Reviewer 3 Report
1. Articles on plant cover must include the characteristics of the plant communities of the study area, even if these plant communities are not part of the later analysis.
Therefore, description of the dominant plant communities in the subchapter „Study area” should be completed (one paragraph or in table). On the other hand, the methodology should explain the degree of detail to which the plant cover is examined. So that there is no doubt.
The conclusions should also point to the direction of extending further research.
2. There are still no websites containing data sources on the reference list. Looking at the references, it is not known what the authors used.
3. Formatting of references should be improved.
4. I leave the number of figures to the Editors' discretion.
Author Response
List of Responses
Dear Editors and Reviewers:
Thank you for your letter and for the reviewers’ comments concerning our manuscript entitled “Spatiotemporal evolution and driving forces of vegetation cover in the Urumqi River basin” (Manuscript ID: ijerph-1949742). Those comments are all valuable and very helpful for revising and improving our paper, as well as the important guiding significance to our research. We have studied the comments carefully and have made corrections which we hope meet with approval. Revised portions are marked in yellow on the paper. The main corrections in the paper and the responses to the reviewer’s comments are as flowing:
Responds to the reviewer’s comments:
- Response to comment: Articles on plant cover must include the characteristics of the plant communities of the study area, even if these plant communities are not part of the later analysis. Therefore, description of the dominant plant communities in the subchapter “Study area” should be completed (one paragraph or in table). On the other hand, the methodology should explain the degree of detail to which the plant cover is examined. So that there is no doubt. The conclusions should also point to the direction of extending further research.
Response: We added a description of the dominant plant communities in lines 110-115 and made the corresponding tables (Appendix table 1) and vegetation distribution maps (Appendix Figure 1). In addition, we submit the table of values of NDVIveg and NDVIsoil in order to interpret the calculated results (Appendix table 2). Finally, in the conclusion, we explain the future direction of the research in lines 654-660.
|
Appendix table 1. Vegetation attribute table. |
||
|
family |
species |
Path ID |
|
Chenopodiaceae |
Haloxylon ammodendron |
1 |
|
Haloxylon Persicum |
2 |
|
|
Ceratoides latens |
3 |
|
|
Camphorosma monspeliaca |
11 |
|
|
Anabasis brevifolia |
4,11 |
|
|
Nanophyton erinaceum |
5 |
|
|
Seriphidium transiliense |
6 |
|
|
Seriphidium borotalense |
7,28 |
|
|
Seriphidium santolinum |
8 |
|
|
Kalidium foliatum |
9 |
|
|
Suaeda dendroides |
10 |
|
|
Gramineae |
Stipa capillata |
12,27 |
|
Poa angustifolia |
13,27 |
|
|
Festuca valesiaca subsp |
14 |
|
|
Festuca ovina |
15,28 |
|
|
Agropyron cristatum |
16 |
|
|
Stipa caucasica |
17 |
|
|
Phragmites australis |
18 |
|
|
Triticum aestivum |
30 |
|
|
Zea mays |
30 |
|
|
Rosaceae |
Rosa sericea Lindl |
19 |
|
Cotoneaster adpressus |
19 |
|
|
Malus pumila Mill |
30 |
|
|
Cyperaceae |
Carex songorica |
20 |
|
Kobresia capillifolia |
21 |
|
|
Carex stenocarpa |
22 |
|
|
Compositae |
Saussurea japonica |
29 |
|
Cremanthodium reniforme |
29 |
|
|
Pinaceae |
Picea schrenkiana |
26 |
|
Ephedraceae |
Ephedra przewalskii |
23 |
|
Tamaricaceae |
Reaumuria songarica |
24 |
|
Leguminosae |
Caragana stenophylla |
25 |
|
Crassulaceae |
Rhodiola rosea |
29 |
|
Cucurbitaceae |
Cucumis melo |
30 |
|
Vitaceae |
Vitis vinifera |
30 |
Appendix Figure 1. Vegetation distribution maps of the Urumqi River basin.
|
Appendix table 2. Values of NDVIveg and NDVIsoil. |
||
|
year |
NDVIv |
NDVIs |
|
2000 |
0.797248 |
0.030334 |
|
2001 |
0.752873 |
0.008891 |
|
2002 |
0.813072 |
0.018738 |
|
2003 |
0.711846 |
0.026076 |
|
2004 |
0.718068 |
0.017333 |
|
2005 |
0.743112 |
0.027248 |
|
2006 |
0.786309 |
0.018308 |
|
2007 |
0.762101 |
0.026306 |
|
2008 |
0.587261 |
0.008304 |
|
2009 |
0.822923 |
0.027764 |
|
2010 |
0.702628 |
0.033905 |
|
2011 |
0.755456 |
0.008804 |
|
2013 |
0.836641 |
0.020191 |
|
2014 |
0.010129 |
0.010129 |
|
2015 |
0.848141 |
0.040466 |
|
2016 |
0.828082 |
0.020238 |
|
2017 |
0.858482 |
0.030346 |
|
2018 |
0.817579 |
0.020223 |
|
2019 |
0.816964 |
0.010122 |
|
2020 |
0.785177 |
0.020173 |
- Response to comment: There are still no websites containing data sources on the reference list. Looking at the references, it is not known what the authors used.
Response: We have made corrections according to the Reviewer’s comments.
- Response to comment: Formatting of references should be improved.
Response: We have made corrections according to the Reviewer’s comments.
Special thanks to you for your good comments.

Reviewer 4 Report
The authors have provided sufficient responses and corrections from the initial review.
Author Response
List of Responses
Dear Editors and Reviewers:
Thank you for your letter and for the reviewers’ comments concerning our manuscript entitled “Spatiotemporal evolution and driving forces of vegetation cover in the Urumqi River basin” (Manuscript ID: ijerph-1949742). Those comments are all valuable and very helpful for revising and improving our paper, as well as the important guiding significance to our research. We have studied the comments carefully and have made corrections which we hope meet with approval. Revised portions are marked in yellow on the paper. The main corrections in the paper and the responses to the reviewer’s comments are as flowing:
Responds to the reviewer’s comments:
- Response to comment: The authors have provided sufficient responses and corrections from the initial review.
Response: We have made corrections according to the Reviewer’s comments.
Special thanks to you for your good comments.